# Feasibility of High-Cellular-Resolution Full-Field, Artificial-Intelligence-Assisted, Real-Time Optical Coherence Tomography in the Evaluation of Vitiligo: A Prospective Longitudinal Follow-Up Study

**DOI:** 10.3390/bioengineering11020196

**Published:** 2024-02-19

**Authors:** Lai-Ying Lu, Yi-Ting Chen, I-Ling Chen, Yu-Chang Shih, Rosalie Tzu-Li Liu, Yi-Jing Lai, Chau Yee Ng

**Affiliations:** 1Department of Dermatology, Chang Gung Memorial Hospital, Linkou Main Branch, Taoyuan 333423, Taiwan; shaihang@hotmail.com (L.-Y.L.); eeehappyrrr@gmail.com (Y.-J.L.); 2Department of Dermatology and Aesthetic Medicine Center, Jen-Ai Hospital, Taichung 412224, Taiwan; 3College of Medicine, Chang Gung University, Taoyuan 333323, Taiwan; 4Apollo Medical Optics, Ltd., Taipei 114, Taiwan; yiting.chen.anny@gmail.com (Y.-T.C.); esther.x10@gmail.com (I.-L.C.); zacheenoao@gmail.com (Y.-C.S.); 5Vitiligo Clinic and Pigment Research Center, Chang Gung Memorial Hospital, Linkou District, New Taipei 33305, Taiwan

**Keywords:** vitiligo, optical coherence tomography, suction blister epidermal grafting surgery, artificial intelligence

## Abstract

Vitiligo, a psychologically distressing pigmentary disorder characterized by white depigmented patches due to melanocyte loss, necessitates non-invasive tools for early detection and treatment response monitoring. High-cellular-resolution full-field optical coherence tomography (CRFF-OCT) is emerging in pigmentary disorder assessment, but its applicability in vitiligo repigmentation after tissue grafting remains unexplored. To investigate the feasibility of CRFF-OCT for evaluating vitiligo lesions following tissue grafting, our investigation involved ten vitiligo patients who underwent suction blister epidermal grafting and laser ablation at a tertiary center between 2021 and 2022. Over a six-month period, clinical features, dermoscopy, and photography data were recorded. Utilizing CRFF-OCT along with artificial intelligence (AI) applications, repigmentation features were captured and analyzed. The CRFF-OCT analysis revealed a distinct dark band in vitiligo lesion skin, indicating melanin loss. Grafted areas exhibited melanocytes with dendrites around the epidermal-dermal junction and hair follicles. CRFF-OCT demonstrated its efficacy in the early detection of melanocyte recovery and accurate melanin quantification. This study introduces CRFF-OCT as a real-time, non-invasive, and in vivo evaluation tool for assessing vitiligo repigmentation, offering valuable insights into pigmentary disorders and treatment responses.

## 1. Introduction

Vitiligo is a common pigmentary disorder characterized by the progressive loss of melanocytes, which manifests as irregular hypopigmented patches, causing substantial degradation in patients’ quality of life and eliciting psychological distress. With a global prevalence of approximately 1–2%, vitiligo’s impact reverberates across diverse populations, warranting comprehensive investigation [1]. Its intricate pathogenesis stems from a complex interplay of genetic and environmental factors, such as monobenzone exposure and mechanical trauma, compounded by T-cell-mediated immune responses and the harmful effects of oxidative stress on melanocytes during the disease’s temporal progression [2]. The evaluation landscape for vitiligo encompasses subjective methodologies such as digital photography, ultraviolet light photography, and various scoring indices like VDAS and VASI, in addition to objective measures like colorimetry, RCM, and image analysis [1,3,4,5,6,7,8,9]. These tools collectively contribute to the characterization and quantification of vitiligo’s clinical presentation.

A paramount consideration in the clinical management of vitiligo is accurately determining disease activity, especially pertinent in pre-operative assessments for interventions like vitiligo tissue grafting. However, a conspicuous void remains concerning the absence of a quantitative, non-invasive, and cellular-level assessment method tailored to the vitiligo pathology’s intricacies. As researchers, we confront the challenge of addressing this deficiency in our diagnostic armamentarium, necessitating the exploration and development of innovative approaches encompassing the disease’s essence and dynamic progression. Pursuing such methodologies is intrinsic to advancing the precision and scope of our understanding of vitiligo, enriching our therapeutic interventions, and bolstering patient care.

Recent strides in skin imaging have ushered in a new era of diagnostic capabilities, propelled by technologies like reflectance confocal microscopy (RCM), optical coherence tomography (OCT), and multiphoton microscopy (MPM). These advancements enable the non-invasive, real-time assessment of histological and cellular intricacies within the skin. While OCT has already established its utility in ophthalmology for diagnosing retinal diseases, its relevance in dermatology has progressively gained momentum. The past few years have witnessed explorations of OCT’s potential in domains including non-melanoma skin cancer, inflammatory skin diseases, nail disorders, vascular conditions, bullous diseases, and melasma [9,10,11,12,13]. Within this milieu, the deployment of RCM for evaluating vitiligo has garnered scholarly attention [14,15,16,17,18]. In contrast, the viability of OCT for vitiligo applications remains relatively unexplored.

Concurrently, the integration of artificial intelligence (AI) into OCT imaging has emerged as a transformative diagnostic tool, particularly in ophthalmology. This synergy has proven invaluable in detecting macular edema, retinal anomalies, and corneal diseases. AI’s discerning precision has, in many instances, surpassed the subjectivity inherent in traditional physician evaluations [19,20,21]. Building on these advancements, our present study is rooted in the dual objectives of probing the feasibility of high-cellular-resolution full-field OCT (CRFF-OCT) in evaluating vitiligo lesions after skin tissue grafting and harnessing the potential of an innovative AI methodology for the quantitative assessment of lesion repigmentation areas and scores. This venture expands the horizons of OCT’s applicability in dermatology and presents a potential paradigm shift in our diagnostic methodologies underpinned by cutting-edge technological collaborations.

## 2. Materials and Methods

### 2.1. Study Design and Participant Recruitment

This pilot study enrolled ten individuals with facial vitiligo who underwent suction blister epidermal grafting (SBEG) and laser ablation at Chang Gung Memorial Hospital, Linkou Branch, between 2021 and 2022. Over six months, detailed clinical assessments were conducted, including disease activity evaluations (VASI score, Vitiligo Disease Activity (VIDA) score, Vitiligo Potential Repigmentation Index (PRI)), dermoscopy, photography, and Wood’s lamp examinations (Table 1). Ethical compliance was maintained with written informed consent, approved by the Institutional Review Board (IRB) of Chang Gung Medical Foundation (IRB: 202002210B0A0).

### 2.2. OCT Imaging Protocol

OCT imaging was performed at specific time points: day 0 (D0), day 30 (D30), day 90 (D90), and day 180 (D180), utilizing the ApolloVue^®^ S100 Image System (Apollo Medical Optics, Ltd., Taipei, Taiwan). This system generated images measuring 897 × 899 pixels, featuring an image resolution of approximately 0.5 μm/pixel, stored with an 8-bit pixel depth. This advanced technology offered a significantly improved resolution, facilitating in-depth skin architecture and dynamics analysis compared to conventional OCT.

### 2.3. In Vivo High-Cellular-Resolution Full-Field Optical Coherence Tomography (OCT)

The ApolloVue^®^ S100 OCT device offered impressive features. With B-scan (vertical) and en face scan (horizontal) functionalities, it achieved resolution levels of 1–2 μm, surpassing the typical vertical and horizontal resolutions of approximately 10 μm and 7.5 μm in conventional OCT. Noteworthy is its provision of vertical images capturing the epidermis, dermis, and dermal–epidermal junction, akin to histopathological characteristics. The respective field of view (FOV) for B-scan and en face ideas measured 500 × 400 μm^2^ (imaging depth) and 500 × 500 μm^2^, enabling comprehensive visualization of the skin microenvironment [22,23].

### 2.4. Artificial-Intelligence-Enhanced Optical Coherence Tomography for Vitiligo Assessment

We harnessed a computer-aided detection (CADe) system with improved AI algorithms to quantify melanin and dendritic cells (DCs) in OCT images from the No. 202002210B0A0 database. Employing cross-sectional reflectance full-field OCT (CRFF-OCT), we sequentially captured face images spanning the skin layers.

This CADe system not only covers the pigment detection function that has been proven effective in pigment disorder research [15], but also adds the function of automatic DC detection. Contrast-limited adaptive histogram equalization (CLAHE) [24] effectively enhanced intensity contrast in localized regions (12.5 × 12.5 µm^2^), enhancing melanin discernibility. We isolated melanin by applying a grayscale threshold above 143 and an area criterion below 843 µm^2^. DCs, characterized by elongated white structures, were identified using the Frangi method [25], excluding results with length < 8.5 µm or area < 33.7 µm^2^. Furthermore, the updated algorithm considered the surrounding information of the detection results to eliminate interferences near structures such as hair, stratum corneum, pigment rings, and skin furrows.

Melanin was sub-categorized into confetti (diameter > 3.3 µm) and granular (diameter of 0.5–3.3 µm) types. Our AI program meticulously scanned OCT images, facilitating the identification of melanin-rich areas and quantified DCs and two types of melanin in terms of number, shape, distribution, and signal intensity, respectively. This pioneering approach expands vitiligo assessment boundaries, enhancing our understanding of underlying dynamics and advancing diagnostic capabilities. The establishment of the ground truth of the artificial intelligence model was scored by two independent readers (CY. Ng and YT. Chen). There are no significant differences in the evaluation results between the two evaluators (*p* value < 0.05). The missing control was avoided by using normal and pathogenic lesions in the same patient for correct comparison.

### 2.5. Statistical Analysis

Descriptive statistics succinctly summarized the demographic and clinical features of the study participants. Continuous variables were presented as means ± standard deviations, while categorical variables were expressed as frequencies and percentages. To evaluate the pigment ratio, we divided the pigment area within the lesion by the corresponding area in normal skin. Prism software (Version 10 GraphPad, San Diego, CA, USA) conducted statistical analyses post data collection and preprocessing. To detect changes in melanin area over time, we used paired sample *t*-tests, comparing melanin areas within the lesion and normal skin regions at basal layers across four-time points: baseline, 30, 90, and 180 days. Significance was defined as a two-sided *p*-value < 0.05.

## 3. Results

### 3.1. Demographic Data and Vitiligo Index Scores

The study encompassed ten participants, with a gender distribution of four males and six females. The average age of the cohort was 34 ± 14.9 years. The mean duration of the last active vitiligo episode was 1.65 ± 1.1 years before the study. Notably, all vitiligo lesions were located on the facial region. The pre-operative evaluation indicated nonactive vitiligo status for all participants, as evidenced using VASI, VIDA, and Vitiligo PRI scores (Table 1 and Table 2).

### 3.2. CRFF-OCT Features of Vitiligo

In the stable phase of vitiligo (D0), CRFF-OCT unveiled a complete loss of melanin at the central region of vitiligo lesions. A month post SBEG grafting (D30), CRFF-OCT imagery displayed multiple highly refractile inflammatory cells alongside melanocytes adorned with prominent dendrites, clustering around the dermal–epidermal junction (DEJ). Conspicuous enhancement in brightness was observed in the papillary dermal rings compared to D0 images. Progressing to the three-month post grafting milestone (D90), there was a discernible reduction in refractile inflammatory cells. Nonetheless, the melanocyte dendrites distributed diffusely near the DEJ persisted. Six months post grafting (D180), a remarkable revival was observed, characterized by a prominent papillary dermal ring, regenerated melanocytes, and robustly developed dendrites (Figure 1).

However, in graft rejection (Figure 2), CRFF-OCT imagery at D30 and D90 displayed an absence of refractile inflammatory cells and dendrite-adorned melanocytes. After a narrowband ultraviolet B phototherapy course, marked clinical improvement and alterations in Wood’s light assessments were noted in the vitiligo patches. Correspondingly, CRFF-OCT images illustrated the emergence of a prominent papillary dermal ring, accompanied by the appearance of newly developed melanocytes and dendrites.

### 3.3. Artificial Intelligence Program for Melanin and Melanocyte Dendritic Cell Detection

In the CRFF-OCT images from the E scan, the automatic scanning of the melanin area and melanocyte dendritic cells is visually represented as green in Figure 3. A noticeable disparity in melanin content and dendritic cell presence was observed between vitiligo and normal skin at D0 (Figure 3). Following one month post SBEG grafting (D30), a substantial increase in melanin and dendritic cells was discernible in comparisons between vitiligo and normal skin (Figure 3). Active dendritic cells were evident in 90% of cases at day 30, while only 50% exhibited clinical repigmentation.

### 3.4. Detection of Meanin and Melanocyte Dendritic Cells Using Artificial Intelligence

The mean total melanin area ratio in CRFF-OCT images exhibited a noteworthy rise from 0.729 (D0) to 1.064 (D180), reflecting statistically significant differences (*p* < 0.05) across each time point follow-up at D0, D30, D90, and D180. Similar trends were evident in the average melanin ratio within the confetti area, escalating from 0.745 (D0) to 1.150 (D180), again demonstrating significant differences (*p* < 0.05). However, the average melanin ratio within the granular area displayed a progression from 0.887 (D0) to 1.260 (D180), with statistical significance observed solely in the D0 and D90 comparison (*p* < 0.05). Collectively, these findings underscored an augmentation in pigmentation post SBEG grafting treatment, with melanin distribution aligning more closely with normal skin attributes (Figure 4A).

A pronounced disparity was noted at D0 (0.634) in the dendritic cell detection between vitiligo and normal skin. Following SBEG grafting, a marked increase in dendritic cells emerged at D30 (1.669), followed by gradual fluctuations approaching the dendritic cell count observed in normal skin by D180 (0.972). This pattern showcased statistically significant differences among the time points D0, D30, D90, and D180 (Figure 4B).

## 4. Discussion

Image analysis has assumed paramount significance in dermatology, revolutionizing diagnostics and treatment evaluation. Incorporating in vivo, non-invasive, real-time repetitive optical coherence tomography (OCT) imaging has proven to be instrumental in various dermatological contexts. Notably, OCT has emerged as a pivotal tool for assessing skin malignancies, encompassing melanoma, basal cell carcinoma, actinic keratosis, and squamous cell carcinoma. It has also demonstrated utility in scrutinizing inflammatory skin disorders like acne, nail psoriasis, and allergic dermatitis [16,26,27,28,29,30].

However, a notable gap persists in the realm of vitiligo evaluation. While OCT has penetrated diverse dermatological arenas, its feasibility and applicability in the intricate domain of vitiligo assessment have yet to be thoroughly explored. This study endeavors to bridge this gap by investigating the hitherto uncharted potential of OCT in the dynamic landscape of vitiligo. Through meticulously exploring OCT’s capabilities, we aspire to expand our understanding of vitiligo’s intricacies and contribute to enhancing diagnostic methodologies and treatment monitoring in this challenging pigmentary disorder.

Our study enlisted a cohort of 10 patients characterized by stable-stage vitiligo. These participants underwent suction blister epidermal grafting (SBEG) vitiligo surgery. Optical coherence tomography (OCT) imaging was employed with an artificial intelligence program for pre-and postoperative evaluations. The study aimed to assess the utility of OCT in characterizing vitiligo lesion dynamics across distinct time points. The comparison of OCT images revealed compelling insights: day 0 exhibited complete melanin loss, and day 30 observed repigmentation accompanied by heightened refractile inflammatory cells. Day 90 unveiled the prominence of melanocyte dendrites and the formation of dermal rings around the dermal–epidermal junction (DEJ), and by day 180, robust dendrites and pronounced dermal rings were evident. This sequential portrayal underscored the potential of OCT in effectively staging vitiligo lesions. Notably, instances of graft rejection post SBEG grafting yielded images devoid of refractile inflammatory cells and melanocyte dendritic cells at D30 and D60. This highlighted the pivotal significance of early detection in grafting success using OCT images. Two dermatologists meticulously assessed all digital photos, and the comparison demonstrated that divergent scorings by different individuals did not exert a discernible impact on the final results (*p* < 0.05).

The realm of vitiligo assessment has long engaged discussions on real-time, repetitive imaging tools, with in vivo reflectance confocal microscopy (RCM) emerging as a prominent contender in these dialogues [31]. However, our study embarks on uncharted terrain, unveiling the potential of optical coherence tomography (OCT) as a pivotal instrument for vitiligo evaluation. This research is a significant stride towards advancing diagnostic methodologies within this domain. Noteworthy findings in our study resonate with the observations reported in the existing literature, particularly the disappearance of papillary rings and the presence of large dendritic melanocytes surrounding hair follicles [11]. These parallels offer valuable insights for comparing vitiligo stability and distinguishing it from other hypopigmented skin conditions [9,32].

OCT devices offer several advantages. The imaging depth extends to the dermal layer, offering transverse and vertical section plane images. The vertical plane of the section, resembling pathology slides, is readily accessible through the B-scan mode. This facilitates swift and repetitive dermatologist evaluations with a minimal learning curve—a contrast to the challenges faced with in vivo real-time RCM. The transverse section of the E-scan mode mirrors RCM, exhibiting visible papillary rings and refractile inflammatory cells. However, employing E mode alone may pose challenges for beginners. It is worth noting that there is no substantive difference in the use of the B-scan or E-scan mode for vitiligo evaluation.

For a comprehensive overview of the pros and cons of RCM and OCT in the context of vitiligo assessment, we have encapsulated this information in Table 3. Through this study, we illuminate the transformative potential of OCT in enhancing our understanding of vitiligo and refining diagnostic approaches in this intricate field.

Our investigation into quantitatively assessing melanin and dendritic cells through artificial intelligence applied to OCT yielded noteworthy findings. Significant differences emerged across successive time points, showcasing a progressive increase in average melanin content within the total and confetti melanin areas (Figure 4A). However, within the granular area evaluations, the lack of significant differences could be attributed to the relatively elevated baseline melanin levels and a smaller sample size. Additionally, an intriguing observation pertains to the conspicuous surge in dendritic cell numbers among melanocytes at D30, underscoring the pivotal role of identifying dendritic cells at this time point through the use of artificial-intelligence-driven OCT images (Figure 4B). Nevertheless, we acknowledge the limitations inherent to our research. The single-center prospective longitudinal design, coupled with a modest sample size, necessitate a careful explanation of our findings.

In summation, OCT emerges as a pragmatic, non-invasive, and swift tool for evaluating vitiligo skin characteristics and tracking surgical treatment responses. The manifestation of repigmentation signs as early as day 30 amplifies its diagnostic utility. As our research progresses, we will focus on determining positive predictive values and establishing correlations with clinical responses, refining our understanding of this dynamic field.

## 5. Conclusions

This study represents a pioneering investigation into the viability of utilizing optical coherence tomography (OCT) in conjunction with artificial intelligence for assessing vitiligo vulgaris post SBEG surgery. In contrast to conventional diagnostic modalities for vitiligo, the application of OCT for skin diagnosis presents an innovative avenue that offers real-time, non-invasive, and in vivo capabilities. This approach holds promise as a quantitative means for evaluating pigmentary disorders, gauging treatment outcomes, and facilitating therapeutic monitoring. Moreover, incorporating artificial intelligence introduces an enhanced dimension of accuracy and efficiency in assessing progression. Through this novel approach, OCT emerges as a potential tool for continually monitoring the evolution of vitiligo, thereby contributing to a comprehensive understanding of the condition and its response to treatment.

## Figures and Tables

**Figure 1 bioengineering-11-00196-f001:**
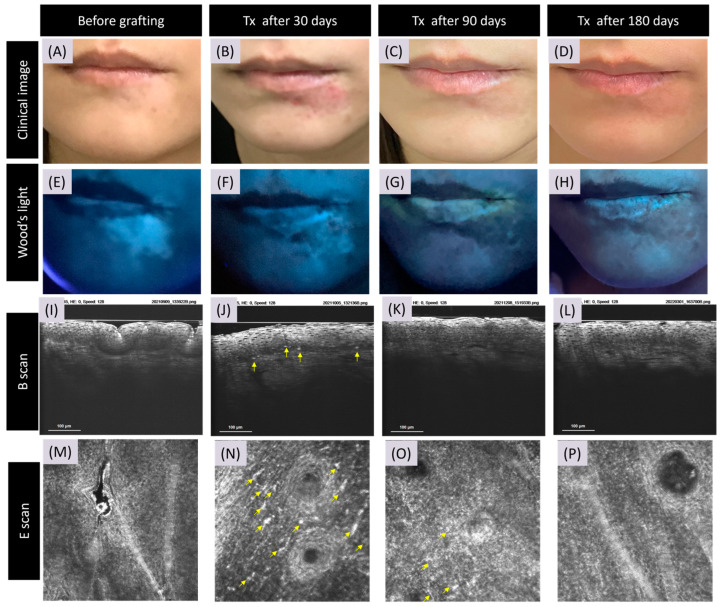
Cellular resolution full-field OCT features of successful skin grafting in vitiligo. The figure illustrates the dynamic transformation of a vitiligo lesion through high-resolution full-field optical coherence tomography (OCT) imaging following suction blister epidermal grafting (SBEG) treatment. The success of the grafting procedure is indicated by notable improvements in melanin content and the presence of active melanocytes at the dermato-epidermal junction (DEJ). Before grafting, the image depicts a facial vitiligo lesion on the jaw (**A**,**E**). The initial state of diminished melanin content beneath the cross-sectional reflection full-field OCT (CRFF-OCT) system is evident (**I**,**M**). After 30 days, the CRFF-OCT reveals the emergence of abundant melanocytes (indicated by ➚ arrow) along the DEJ (**B**,**F**,**J**,**N**). Progressing to 90 and 180 days post grafting, the pigmentation of the lesion area approaches parity with normal skin pigmentation (**C**,**D**,**G**,**H**), and the resurgent melanin at the DEJ becomes notably pronounced (**K**,**L**,**O**,**P**). Note: Imaging scans are denoted as transverse sections (E scan) and vertical sections (B scan), respectively, showcasing the lateral and depth perspectives of the observed changes.

**Figure 2 bioengineering-11-00196-f002:**
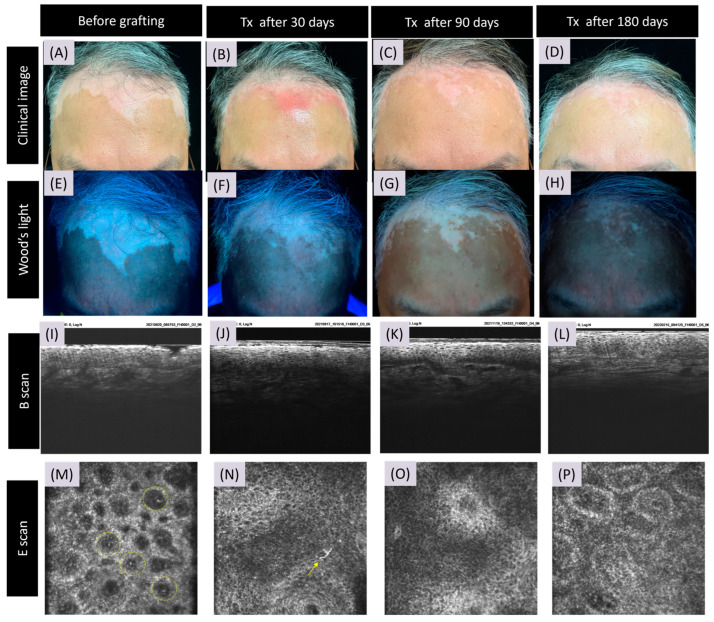
Cellular-resolution full-field oct features of rejected skin grafting in vitiligo. This figure portrays the effects of skin grafting on a vitiligo lesion through the lens of cellular-resolution full-field optical coherence tomography (CRFF-OCT). The presented case demonstrates a grafting procedure resulting in reduced melanin content—a situation categorized as a rejected case—accompanied by immunocytes along the dermato-epidermal junction (DEJ). Before grafting, the image portrays a vitiligo lesion situated upon the forehead. (**A**,**E**). The CRFF-OCT reveals the presence of highly refractive inflammatory cells within the papillary dermis (**I**,**M**). Following 30 days, the CRFF-OCT exposes a limited number of melanocytes (highlighted by the ➚ arrow) at the DEJ (**B**,**F**,**J**,**N**). Progressing to 90 days post grafting, the pigmentation restoration at the lesion site indicates a rejected graft (**K**,**O**). After treatment with azathioprine 50 mg/day for 90 days, repigmentation with papillary rings, melanocytes, and melanin deposition was found (**L**,**P**). Also, between days 90 and 180 post-grafting, notable repigmentation occurred in the lesion during azathioprine administration, (**C**,**D**,**G**,**H**).

**Figure 3 bioengineering-11-00196-f003:**
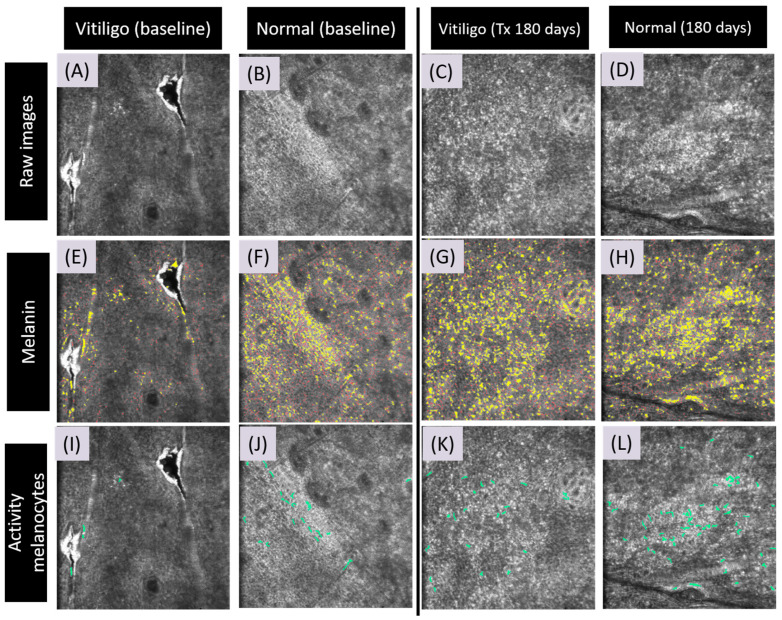
Application of artificial intelligence for detecting melanin and dendritic cells in E-scan OCT imaging. This figure showcases the utilization of artificial intelligence (AI) for identifying and analyzing melanin and dendritic cells in optical coherence tomography (OCT) imaging using E-scan data. **Raw Images:** The provided images depict the original CRFF-OCT saved images (**A**–**D**). Melanin Detection: A discernible discrepancy in melanin content becomes apparent when contrasting the baseline images, which encompass both normal skin and vitiligo samples (**E**,**F**). Specifically, a heightened melanin presence is noticeable and highlighted within the boundaries of normal skin. Following six months of therapeutic intervention, a visible convergence in pigment content towards that of normal skin is discernible (**G**,**H**). Activity Melanocytes Detection: A comparison of the baseline image, including normal skin and vitiligo samples, illustrates that vitiligo images initially exhibit fewer activated melanocytes (**I**,**J**). After six months of treatment, a corresponding increase in the count of activated melanocytes in vitiligo images, mirroring that observed in normal skin, becomes evident (**K**,**L**).

**Figure 4 bioengineering-11-00196-f004:**
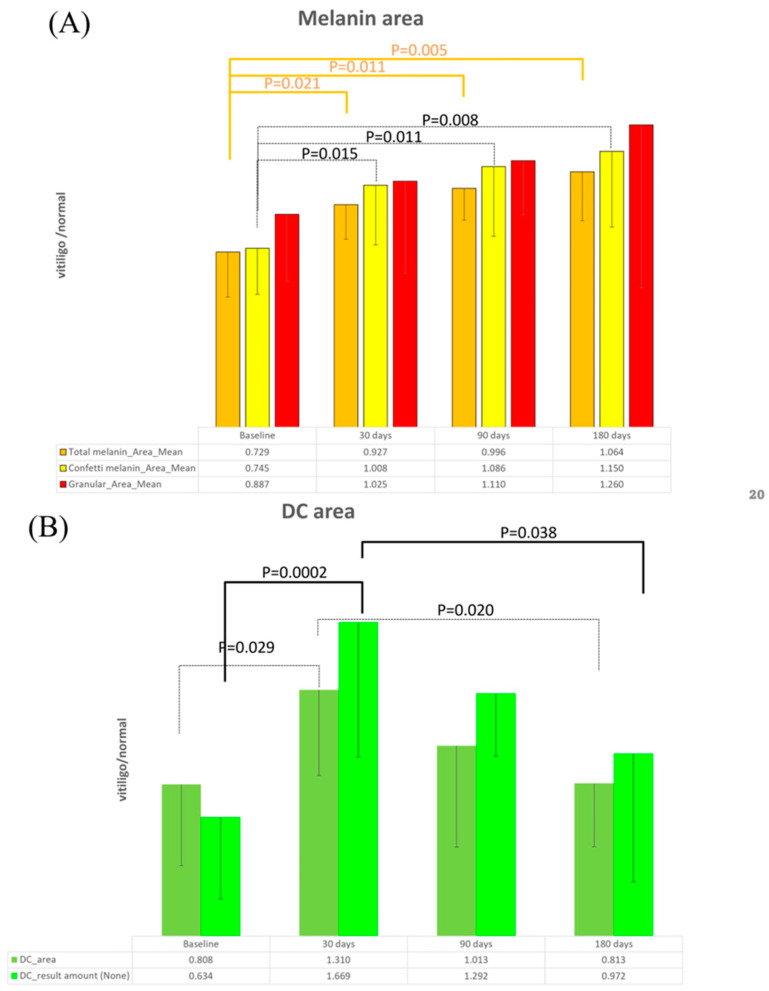
Detection of melanin and melanocyte dendritic cells using artificial intelligence application. (**A**) Post treatment analysis: AI-driven transformation and depigmentation. Following treatment, the application of artificial intelligence (AI) for analysis—using normal skin as a reference—reveals a discernible trend wherein vitiligo-affected areas progressively resemble the characteristics of normal skin. Notably, OCT imaging substantiates its capacity to capture the depigmentation phenomenon accurately. The initial assessment establishes a vitiligo-pigment-to-normal-skin-pigment ratio of 0.729. (**B**) Activated melanocyte detection: advancements After 30 Days. In detecting activated melanocytes, a more extensive area and an elevated count of melanocytes are evident after a 30-day treatment period.

**Table 1 bioengineering-11-00196-t001:** Demographics and clinical characteristics of patients undergoing epidermal grafting.

Characteristics	N = 10
Gender (male/female)	4/6
Age (years; mean + SD)	34 + 14.9
Recipient sites	
- Eyebrow	2
- Jaw	3
- Forehead	2
- Postauricular area	1
- Periorbital	2
VIDA score at first visit	
- 0	2
- −1	8
Average duration of active vitiligo (years; mean + SD)	1.65 ± 1.1
VASI score at first visit	
- 100% (complete depigmentation)	7
- 90% (specks of pigment present)	3
Vitiligo PRI score at first visit	
- Type A *	1
- Type C *	9

Abbreviations: VASI—Vitiligo Area Severity Index; VIDA—Vitiligo Disease Activity score; PRI—Potential Repigmentation Index; * Type A: Perifollicular Repigmentation and Residual Melanocyte in Follicle; Type C: Leukotrichia; Definition of repigmentation: pigmentation exceeding 75%.

**Table 2 bioengineering-11-00196-t002:** Longitudinal evaluation of grade of repigmentation in vitiligo lesions after epidermal grafting.

	Sex/Age	Last Time of Active Vitiligo	Grafting Area	VASI Score *			
				V1	V2	V3	V4
**1**	F/37	>1 year	Eyebrow and forehead	100%	25%	25%	10%
**2**	F/20	1 year ago	Jaw	90%	25%	25%	10%
**3**	F/55	1 year ago	Scalp	100%	25%	25%	25%
**4**	M/24	1 year ago	Postauricular area	100%	25%	25%	10%
**5**	F/65	>1 year	Forehead	90%	75%	10%	25%
**6**	M/24	2 years	Eyebrow	100%	25%	25%	10%
**7**	M/39	2 years	Jaw	100%	50%	10%	10%
**8**	F/33	4–5 years	Jaw	90%	50%	10%	10%
**9**	M/44	2 years	Periorbital	100%	10%	10%	10%
**10**	F/35	>1 year	Periorbital	100%	10%	10%	10%

* VASI score: Vitiligo Area Severity Index: (100%: complete depigmentation; 90%—specks of pigment present; 75%—depigmented area exceeds the pigmented area; 50%—pigmented and depigmented areas are equal; 25%—pigmented area exceeds depigmented area; and 10%—only specks of depigmentation present).

**Table 3 bioengineering-11-00196-t003:** Comparison of vitiligo evaluation tools between RCM and OCT.

	Reflectance Confocal Microscopy (RCM)	Optical Coherence Tomography (OCT)
**Devices and specifications**	Lucid Vivascope 1500^®^ (Henrietta, New York, NY, USA)	Apollo Medical Optics Inc.
**Advantages**	- Non-invasive, real-time examination of skin - Achieves cellular-level resolution	- Non-invasive, real-time examination of skin - Achieves cellular-level resolution - Provides depth of skin imaging to the dermal layer - Shorter learning curve for proficiency
**Imaging section plane**	- Offers imaging in transverse section plane	- Offers imaging in both transverse and vertical section plane
**Limitations**	- Depth of view limited to the papillary dermis - Presents primarily horizontal section view - Longer learning curve for interpretation	- Provides a restricted field of vision, requiring broader coverage
**Score index and definition**	RCM scoring: (1) Pigmentation status: +1 for presence of remaining pigment −1 for complete loss of pigment (2) Border status of vitiligo lesions: +1 for indistinct border −1 for clear border (3) Inflammatory cell infiltrate: +1 for presence of inflammatory cells −1 for absence of inflammatory cells (4) Melanocyte regeneration −1: if dendritic melanocytes appear in vitiligo lesion Total score interpretation: - Stable stage: Total score < 1 - Active stage: Total score > 1 - Rapid active stage: Total score > 2	OCT scoring: OCT melanin grading is determined by the pigment area ratio score at the junction of the dermis and epidermis, categorized as follows: (1) Grade 1: 0–25% (2) Grade 2: 26–50% (3) Grade 3: 51–75% (4) Grade 4: 76–100%

## Data Availability

Access to the dataset utilized in this study is attainable upon request. For data access and further inquiries, please feel free to reach out to the corresponding author, Chau Yee Ng, at mdcharlene@gmail.com.

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
