# Peer review of "Feasibility of High-Cellular-Resolution Full-Field, Artificial-Intelligence-Assisted, Real-Time Optical Coherence Tomography in the Evaluation of Vitiligo: A Prospective Longitudinal Follow-Up Study"

_bioengineering, 2024, doi:10.3390/bioengineering11020196_

Round 1
Reviewer 1 Report
Comments and Suggestions for Authors
This study introduces CRFF-OCT as a real-time, noninvasive, and in vivo evaluation tool for assessing vitiligo repigmentation. This study is well-developed, well-written and well-organized, which only needs a minor revision before publication.
1. More references should be cited support the statements in the introduction. Specifically, there are lots of techniques listed in Line 42-58, which, however, don’t have references to support them.
2. Table 2: The format should be Sex/Age. However, the authors use format as Age/Sex.
3. The captions of the figures should be provided under the figures. The current version can cause misunderstanding in reading.
Reviewer 2 Report
Comments and Suggestions for Authors
The work presents a case of Optical Coherence Tomography used in the detection, assessment, and treatment monitoring of vitiligo disease. The dual purpose of the paper is to, a) present the non-invasive technique and quantify its success, and, b) utilize novel, cross-cutting tools of AI in the specific field of bio-medical research. Two sentences from the paper introduction showcase this duality:
"...integrating Artificial Intelligence (AI) into OCT imaging has emerged as a transformative diagnostic tool..."
and
"[B]uilding on these advancements, our present study is rooted in the dual objectives of probing the feasibility of High Cellular Resolution Full-Field OCT (CRFF-OCT) in evaluating vitiligo lesions after skin tissue grafting and harnessing the potential of an innovative AI methodology for the quantitative assessment of lesion repigmentation areas and scores."
The paper succeeds in the first task, i.e. of presenting the OCT methodology, as well as discussing and quantifying its success, albeit utilizing a rather small cohort/sample. Some more work is required before the second task is considered complete. The use of AI as a 'black box' is opening up basic epistomological questions in many fields of research these days, and this work falls right into the same category. The authors make a convincing case about their methodology and its - qualified - success, but they need to clarify the contributions of AI models into that success. in many respects this is the prototypical case of "missing controls", i.e. not contrasting what happens with and without use of AI in their technique. One expects the use of such technology to improve the speed and/or the sensitivity of the technique, but the case must be made by the authors as to which one (or maybe both?) of these is taking place here.
Of course, it is a tall order to request a full account of how these novel tools are operating, since this discussion seems to be at the same in and above everybody's mind these days. But at least some comments must be inserted into the paper, to clarify the issue for the readers. While the present paper works as a validation and verification of the optical technique and the associated methodology, some control/validation is also need for use of the AI tool itself.
Reviewer 3 Report
Comments and Suggestions for Authors
Dear authors,
our study was very interesting and usefully to improve vitiligo diagnosis.
I have some suggestions to ameliore this study:
I think that statistical section would be extend. You can analyzed demographical and clinicopathological feauteres of ten pazients between AI results.
You can extend Reference section.
Round 2
Reviewer 2 Report
Comments and Suggestions for Authors
The authors addressed the reviewer's comments about the AI component of their work, to the extent possible when describing these tools, and their paper can be published in its present form.
